# The Synergy of Water Resource Agglomeration and Innovative Conservation Technologies on Provincial and Regional Water Usage Efficiency in China: A Super SBM-DEA Approach

Rizwana Yasmeen [1,2] , Gang Hao [3], Yusen Ye [4], Wasi Ul Hassan Shah [5,*] and Caihong Tang [1]

1 School of Economics and Management, Panzhihua University, Panzhihua 617000, China; rizwana_239@yahoo.com (R.Y.); tangcaihong-florence@foxmail.com (C.T.)
2 Department of Economics, University of Religions and Denominations, Qom 37491-13357, Iran
3 Department of Management Science, City University of Hong Kong, Hong Kong 999077, China; msghao@cityu.edu.hk
4 Institute for Disaster Management and Reconstruction, Sichuan University, Chengdu 610017, China; moriye@scu.edu.cn
5 School of Management, Zhejiang Shuren University, Hangzhou 310015, China
* Correspondence: wasi450@yahoo.com

**Abstract:** China is currently facing the significant task of effectively managing its water resources to satisfy the rising needs while grappling with the growing worries of water shortage. In this context, it becomes crucial to comprehend the importance of resource agglomeration and technological adoption. Thus, this research examines the relationship between water resource agglomeration and the adoption of innovative conservation technologies in enhancing water usage efficiency at provincial and regional levels in China (2006–2020). In the first stage, the study utilizes a super SBM-Data Envelopment Analysis (DEA) methodology to evaluate the water usage efficiency of China's provinces and regions. In the second stage, we find the dynamic nexuses between water resources, water technologies (recycling, sprinkler irrigation) and water usage efficiency by applying a systematic econometric approach. SBM-DEA analysis revealed that Beijing (1.08), Shaanxi (1.01), Shanghai (1.23) and Tianjin (1.01) remained the higher efficient over the years. Six provinces (Guangdong, Shandong, Jiangsu, Inner Mongolia, Hebei, and Zhejiang) are in the middle ranges (0.55–0.83). In contrast, nineteen provinces have the lowest water usage efficiency (0.21–049). Qinghai and Ningxia are on the lowest rank (0.21) and (0.22), respectively. The findings recommended that the water resources impact is negative. In comparison, the impact of water-saving mechanisms on the efficiency of water usage seems to be positive, as recycling technology significantly enhances the water usage efficiency in China's province. The study found that GDP growth has a negative impact on water usage efficiency in the early stages of economic development. Still, as economies mature, this negative impact diminishes, indicating a tendency to allocate more resources to water conservation and efficiency. Water recycling technology, the modernization of irrigation methods, and water resource management can enhance water efficiency.

**Keywords:** water resources; water technologies; water usage efficiency; SBM-DEA

## 1. Introduction

The efficiency of water usage in the economy is a complex and essential factor that supports a country's sustainable development and environmental responsibility [1]. The index quantifies the efficiency with which a nation employs its water resources to sustain its economic endeavors, including many sectors like agriculture, industry, energy generation, and domestic usage. A greater water usage efficiency index signifies that the economy can generate more goods and services while utilizing smaller water resources [2]. Water usage efficiency offers economic benefits and ecological responsibility by alleviating pressure on

limited water supplies and addressing the negative environmental consequences linked to excessive water consumption, including the depletion of groundwater and the deterioration of aquatic environments [3].

China has also decided to improve its water utilization efficiency in this situation. China, the most populated country globally and exhibiting rapid economic growth, is confronted with an escalating need for water resources to sustain its industrial sector, agricultural activities, and urban areas [4]. Differences between supply and demand, unequal dispersion of water resources, continuing flooding and waterlogged soil incidents, agricultural expansion [5], and water conservation issues remain significant challenges in China's water management landscape. Concurrently, there has been a growing emphasis on national objectives regarding water scarcity, pollution, and the degradation of rivers and lakes. The convergence of these factors, namely climate change and pollution, has resulted in a scenario wherein water scarcity progressively assumes a sense of urgency. China's gross domestic product per water unit is considerably lower than the global average [6]. On a global scale, the average GDP per cubic meter of water is approximately USD 36, whereas China's GDP per cubic meter is roughly USD 3.50 [7]. China possesses several significant rivers in the South Asian region, yet the distribution of water resources throughout time and space exhibits notable disparities. Out of the total of 31 administrative regions located on the mainland, it is observed that eight regions are experiencing a substantial shortfall in water resources, while the remaining 20 regions are encountering a relatively minor scarcity of water. These water-scarce provinces have significant difficulties in all aspects of water management, including agriculture, industry, households, and even drinking water [8]. These issues posed by a lack of accessible water will ultimately impact people's health and day-to-day lives and impede progress toward achieving sustainable growth. Furthermore, the limited level of production technology in domestic, industrial, and agricultural contexts leads to the squandering of resources and the poor utilization of water [9]. So, China has consistently incorporated water conservation goals and targets into its Five-Year Plans (FYP). Such as a new China's 14th Five-Year Plan (covering 2021–2025) plan was unveiled in January 2022. The goal of this plan is to substantially augment China's capacity for national water security by 2025 through four primary points: (i) enhancing the capacity to mitigate floods and droughts, (ii) improving the capacity to conserve water resources, (iii) enhancing the capacity for managing water resources and optimizing allocation, and (iv) fortifying the ecological protection and governance of major rivers and lakes. The FYP also provides recommendations for mitigating agricultural water consumption. Consequently, in the next quinquennium, China intends to advance reforms in crucial domains of water conservation, enhance the innovative advancement of water conservation, and modernize the water management framework through the implementation of a nationwide water-saving campaign and smart water network, alongside the execution of significant water infrastructure projects.

Therefore, it would be right to say that China's effective management and efficient utilization of water resources have emerged as significant focal points. Within this environment, two primary tactics have surfaced as fundamental pillars to maximize water usage efficiency [10]. These strategies encompass the agglomeration of water resources and the implementation of innovative conservation technology.

Resource agglomeration pertains to the strategic development of industries and metropolitan centres intending to maximize the efficient usage of water resources [11]. By strategically clustering industry and urban centres in particular geographical areas, it is possible to enhance the effective allocation of water resources and optimize the infrastructure for water treatment and distribution. This practice results in a decrease in the overall burden on water resources and a waste reduction, significantly contributing to achieving more sustainable water management.

Moreover, adopting technology plays a crucial role in tackling the water difficulties faced by China [12]. The use of advanced water treatment technologies, including desalination, wastewater recycling, and efficient irrigation systems, can significantly improve the

overall efficiency of water utilization. Smart water management systems, which leverage data analytics and sensor technologies, have the potential to facilitate real-time monitoring of water quality and consumption patterns [13]. This capability allows for a more agile and efficient allocation of resources, leading to improved outcomes in resource management.

The interaction between the concentration of resources and the use of technology is of utmost importance [14]. The facilitation of modern water-saving technology can be enhanced by the intentional clustering of industry and cities, resulting in a synergistic effect that optimizes water resource efficiency [15,16]. Furthermore, allocating resources towards advancing research and development in establishing water technologies can foster economic expansion while tackling the pressing issue of water scarcity. A comprehensive understanding of the complex interplay between resource concentration and technological assimilation is paramount in pursuing sustainable water resource management in China. The advanced water-saving approach could help the nation manage complex water scarcity concerns. By doing so, it can guarantee the provision of clean water to its populace while preserving the environment for the benefit of future generations. Furthermore, this could also serve as a model for other regions facing similar water resource issues in a water-scarce world.

Innovations in water-efficient manufacturing, irrigation, and water recycling have a major impact. Furthermore, implementing policy measures, such as enforcing rigorous water quality rules, establishing water pricing mechanisms, and providing incentives to encourage water-saving techniques, can facilitate efficiency enhancements [17]. Monitoring water usage efficiency over time offers politicians, entrepreneurs, and researchers' valuable insights into the capacity of an economy to effectively manage the trade-off between economic development and environmental preservation [18]. This statement underscores the efficacy of tactics employed in managing water resources, identifies areas that require enhancement, and facilitates the assessment of the consequences of different initiatives. Optimizing water usage efficiency is an issue of economic prudence and moral and environmental responsibility in a time of rising water scarcity concerns, unpredictable weather, and ecological pressure [19]. This aids in the assurance that nations can fulfil the needs of their populations and industries while preserving this invaluable resource for both current and future generations. Therefore, it is imperative for governments, corporations, and communities to persistently prioritize the augmentation of water use efficiency as an essential element of their strategy for sustainable development.

The above and introductory discussion showed that water-related problem has gained world attention; correspondingly, China is also entangling the situation and taking measures regarding water resources management (Three Red Lines water policy). Despite the ongoing work in different domains, wide-ranging literature is still silent on water technology and resources' role in enhancing water usage efficiency. Secondly, this study differs as it captures water technology's role in managing the resources efficiently and improving water usage efficiency in China. Third, education incorporation distinguishes this study from others as it can be an optimizing source to increase water usage efficiency. To this end, this study is developed to explore the impact of water resource agglomeration and innovative conservation technologies on the water usage efficiency at the province and regional levels in China from (2006–2020). This research would provide insight into the relationship between resource agglomeration and technology adoption. By doing so, it seeks to contribute to understanding how economic growth and environmental sustainability can be achieved. This study further contributed in the following ways: First, this study has two estimation stages. In the first stage, the water usage efficiency of the provinces and regions is measured through the super SBM-DEA approach. In the second stage, we find the dynamic nexuses between water resources, water technologies and water usage efficiency by applying a systematic econometric approach. This would show how provincial and central governments can use the resources and technologies to improve their water efficiency. Further, it will highlight how the conversion process can lower the inputs (e.g., water, labor, and capital) to produce higher output with fewer toxins. Third, we

capture the nonlinear growth behavior concerning water usage efficiency by incorporating the first and second stages of growth impacts. Fourth, we assess the education impact on increasing water usage efficiency. Lastly, we incorporated the interaction (moderating) impact of advanced technology and water use methods with water resources to increase water usage efficiency.

## 2. Literature Review

Water, commonly known as the "blue gold" of our world, is an indispensable and essential resource that supports life, agriculture, industry, and the preservation of the environment [20]. With the rise in populations, the expansion of industries, and the impact of climate change on water availability, the need to prioritize effective and sustainable water management becomes of utmost importance [21].

Water usage efficiency will shape a more sustainable future as the globe faces climate change and rising water needs. Therefore, several researchers analyze water usage efficiency and its influencing factors in various contexts. In one strand of literature, scholars focused on water usage efficiency. For example, Xu et al. (2021) [22] investigate agricultural water usage efficiency in China, where agriculture represents the most significant water consumer. The study indicates a notable rebound effect despite the anticipated benefits of enhanced agricultural water use efficiency in mitigating water scarcity. This rebound effect undermines the anticipated water conservation outcomes. The theory (rebound effect, also known as the Jevons paradox) suggests that individuals or organizations, upon achieving enhanced resource efficiency, such as in the case of water consumption, may tend to increase their resource utilization due to its reduced cost or increased accessibility.

Consequently, this behavior counteracts the initial benefits derived from the improved efficiency. Further, the results show that water efficiency in agriculture has a negative relationship with total water use. However, the magnitude of this correlation differs among regions. The study emphasizes the importance of effectively controlling the size of irrigation systems following the existing water resources while concurrently enhancing water utilization efficiency in agricultural practices. According to Callejas Moncaleano et al. (2021) [23], a reason for the greater-than-anticipated demand can be attributed to the inefficient utilization of water resources. They claimed that the advancement towards attaining water use efficiency is experiencing a sluggish pace, particularly in numerous developing nations characterized by the significant deterioration of natural resources, sluggish economic expansion, and an absence of robust institutions to coordinate efforts effectively. Human behavior is identified as a contributing factor to the inefficiency of water usage. Lu (2019) [24] analyzes the relationship between industrial water use efficiency and the environment. They found that water conservation can reduce carbon emissions.

Guerrini et al. (2013) [25] article investigates the Italian water sector's economies of scale, scope, and density. The authors argued that public utilities may boost water sector efficiency by pursuing policies like expanding their operations, diversifying their investment portfolios, and giving preference to areas with high population densities. Hatamkhani and Moridi (2021) [26] also highlighted the limitation of freshwater resources and the growing demand for water by studying an integrated water allocation model that combines economic and social aspects affecting water supply and demand. They applied a reliability-based multi-objective optimization–simulation approach. In another study, Hatamkhani et al. (2020) [27] developed a simulation—optimization model to study the optimal design of the hydropower reservoirs in maximizing the energy generation and minimizing the flood damage.

The second strand of literature focused on water resource management technologies or efficiency. For instance, Qiao et al. (2020) [28] examine water technology economics in the context of water shortage. The research reveals that water technologies boost GDP growth. Additionally, water science and technology breakthroughs drive all water-related innovations. Further argued that water technical efficiency affects economic growth differently by area, depending on local water governance. This study emphasizes the

necessity of research, innovation, and effective governance for sustainable water resource utilization in Northwest China and the potential of water technology to boost economic growth in water-scarce locations. Yang et al. (2022) [29] examine the effects of advancements in water technology on water conservation in China. The study demonstrates that the impact of technical advancement differs among locations in China, with the industrial composition exerting a notable influence on the reduction of water usage.

Qiao et al. (2022) [30] investigate the nexuses between water technology and sustainable development. They found that using water technology enhances the significance of water resources with economic development. The study posits that policy-driven measures can facilitate water technology development in water-scarce emerging nations in the short term; however, long-term advancements are primarily driven by changes in pricing. Similarly, some other studies use technology factors to evaluate water efficiency, resulting in technological progress (and technical efficiency) driving freshwater total factor productivity [31–33]. Ji and Wang. (2015) [34] discovered that technical progress significantly improves total factor productivity in China's freshwater utilization efficiency. Molinos-Senante's [35] study on water usage in England shows that productivity increased with technological improvement.

To sum up, we found scant literature on the agglomeration of water technology in terms of recycling wastewater, sprinkler irrigation and water reservoirs with water resources to enhance water usage efficiency. Additionally, economic growth and education impact make this study more comprehensive in the literature. Third, economic growth has different stages and has different economic consequences. This study incorporated the first (primarily attributed to the scale effect of production growth) and second stage of economic growth (mostly attributed to technique effect) to assess the water usage efficiency. Finally, it would be a valuable addition, as this study differs as it captures water technology's role in managing the resources efficiently and improving the water usage efficiency in China.

## 3. Materials and Methods

Water scarcity poses a significant global challenge in the contemporary era. Effective water management involves strategically arranging water supply and treatment facilities for optimal performance [36]. China faces various water-related challenges, including imbalances between water supply and demand, inequitable distribution of water resources, and the imperative need for water conservation [37]. Further, water scarcity poses a significant challenge to agriculture's sustainable growth and national food security. Besides, China's economic expansion has resulted in a significant rise in both home and industrial wastewater, thereby playing a crucial role in the degradation of environmental conditions. Therefore, technology adoption can positively affect water usage efficiency. Regions that adopt innovative conservation technologies will likely have improved water resources actively. Further, it is expected that regions with more concentrated and accessible water resources will have better Water usage efficiency. Therefore, the study evaluates the agglomeration of water resources and water innovative methods (technology) impact on water usage efficiency from (2006–2020). This study has two estimation stages. First, the study estimates the water usage efficiency of 29 provinces and regions (Table A1, see Appendix A). Subsequently, the study applies the econometric approach to assess the connection between the dynamic relation of the concerned parameters.

The primary empirical models are composed as follows:

$$WUEF_{it} = \alpha_0 + \alpha_1 WRS_{it} + \alpha_2 GDP_{it} + \alpha_3 GDP^2_{it} + \alpha_4 Sind_{it} + \alpha_5 EU_{it} + \\ \alpha_6 pop_{it} + \alpha_7 \mu_{it} \tag{1}$$

$$WUEF_{it} = \alpha_0 + \alpha_1 WR_{it} + \alpha_2 GDP_{it} + \alpha_3 GDP^2_{it} + \alpha_4 SPR_{it} + \alpha_5 WTRS_{it} + \\ \alpha_6 X_{it} + \alpha_8 \mu_{it} \tag{2}$$

Equation (1) uses key control parameters to explain how water resources affect water usage efficiency. $i \ldots, N$ are the provinces and $t$ = period. The $WUEF$ is water usage

efficiency, and *WRS* indicates water resources of province *i* in year *t*. *GDP* (Per capita gross domestic product) to represent the initial level of growth, $GDP^2$ is square *GDP* to assess the second phase of development impact. *Sind* (Secondary Industry), *EU* (secondary education), *pop* are control parameters. The subsequent equation uses water recycling (*WR*) and sprinkler irrigation (*SPR*) as technology factors. *WTRS* is the water reservoir of the *i* province. *X* are the control parameters as Equation (1).

A synergy can be created between water resources, water-conserving technologies, structure and water use. Technology can trigger the function of water consumption efficiently. We include the interaction term in Equation (3) to evaluate the impact of recycling, sprinkle irrigation methods, and reservoirs with water resources on water usage efficiency. $\mu_{it}$ is the error term.

$$WUEF_{it} = \alpha_0 + \alpha_1 WRS_{it} + \alpha_2 GDP_{it} + \alpha_3 GDP^2_{it} + \alpha_4 WR_{it} + \alpha_5 SPR_{it} \\ + \alpha_6 WTRS_{it} + \alpha_7 INTR_{it} + \alpha_8 X_{it} + \alpha_9 \mu_{it} \tag{3}$$

where's, *INTR* are interaction terms, i.e., $(WRS \times WR)$, $(WRS \times SPR)$ and $WRS \times WTRS$. Detailed variable descriptions and data sources are given in Table A2.

### 3.1. Empirical Methods

This study follows two main estimation stages. In the first stage, we calculated the water usage efficiency by super SBM-DEA [38–41]. In the second stage, we apply various economic approaches to empirically assess China's water conservation technologies and water usage efficiency. The empirical path followed is given in Figure 1.

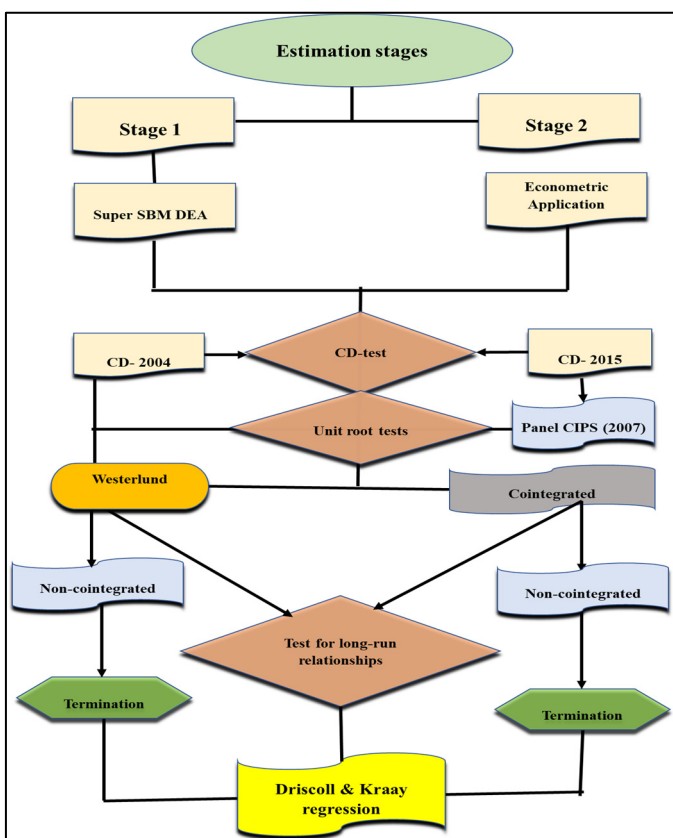

**Figure 1.** Flow-chart of estimation procedure.

Super SBM Data Envelopment

Tone's 2002 [42] introduction of the Super-Efficiency Slack-Based Measure (SBM) model expands the conventional Data Envelopment Analysis (DEA) paradigm by assessing DMU efficiency based on input and output parameters simultaneously. The typical

radial DEA model does not include slack variables, but the Super-Efficiency SBM model does, allowing for a more thorough evaluation of DMUs. Incorporating slack variables solves the radial model's shortcomings, allowing effective DMUs differentiation. The Undesirable Super-SBM model, introduced by Tone's work in 2003 [43], is a game-changing development considering adverse outputs and rates efficient units. The efficiency analysis community may thank this model for pushing the discipline forward.

If there are n DMU and each one has m inputs, then s 1 and s 2 are, respectively, the good and bad outputs. The input-output matrix has the formulas $X = [x_1 \cdots x_n] \in R^{m \times n}$, $Y^{nd} = \left[ y_1^d \cdots y_n^d \right] \in R^{s_1 \times n}$, and $Y^u = \left[ y_1^A \cdots y_n^{ut} \right] \in R^{s_2 \times n}$. Below is the expression of the super-efficient SBM model with bad output.

$$\rho^* = \frac{\frac{1}{m} \sum_{i=1}^{m} \left( \frac{\bar{x}}{x_{ik}} \right)}{\frac{1}{(s_1 + s_2)} \left( \sum_{r=1}^{s_1} \frac{\overline{y^d}}{y_{rk}^d} + \sum_{t=1}^{s_2} \frac{\overline{y^u}}{y_{rk}^u} \right)} \tag{4}$$

$$\text{s.t.} \begin{cases} \bar{x} \geq \sum\limits_{j=1,4k}^{n} x_{ij} \lambda_j; i = 1, 2, \cdots m \\ \overline{y^d} \leq \sum\limits_{j=1, \neq k}^{n} y_{rj}^d \lambda_j; r = 1, \cdots, s_1 \\ \overline{y^{u\mu}} \geq \sum\limits_{j=1, \neq k}^{n} y_{tj}^u \lambda_j; t = 1, \cdots, s_2 \\ \lambda_j \geq 0, j = 1, 2, \cdots n, j \neq 0 \\ \bar{x} \geq x_{ik}; y^d \leq y_{rk}^d; \overline{y^\mu} \geq y_{dk}^u \end{cases}$$

The slack variables of input, desirable output, and undesirable output, respectively, are $\bar{x}, \overline{y^d}$ and $\bar{y}^u$ in the formula; $\lambda j$ is the weight vector; and $\rho^*$ is the model's optimal solution when $\rho^* \geq 1$, the DMU is effective.

### 3.2. Econometric Strategy

This panel-based study uses cross-dependence, stationarity, and long-run effects for reliable findings.

### 3.2.1. Evaluating Cross-Dependence and Unit Root

When undertaking panel studies, it is imperative to consider the potential presence of cross-sectional dependency. The phenomenon in question occurs when the residuals of panel regression models are affected by shared, latent disturbances. Neglecting this interdependence might result in inconsistent estimations and erroneous inferences when employing conventional estimation techniques. To effectively tackle this concern and mitigate the occurrence of model misspecification, we utilize the cross-sectional dependency (CD) analysis technique pioneered by Pesaran [44]. This methodology not only helps ensure the precision of our results but also tackles the issue of size distortion. Significantly, the CD test demonstrates applicability to a broad spectrum of models, encompassing stationary dynamics and unit root heterogeneous panels. This holds even in scenarios where the time series dimension (T) is limited, and the cross-sectional dimension (N) is substantial.

The Pesaran's CD Statistic is as follows:

$$CD = \sqrt{\frac{2T}{N(N-1)}} \left( \sum_{i=1}^{N-1} \sum_{j=i+1}^{N} \hat{\rho}_{ij} \right) \tag{5}$$

where $\hat{\rho}_{ij}$ is the sample estimate of the pair-wise correlation of the residuals.

The subsequent crucial task entails ascertaining the order of integration of the variables. Including this phase is crucial to address the issue of cross-sectional dependence within the panels. When considering this matter, it is important to acknowledge that conventional unit root tests, namely first-generation unit root tests like those suggested by Phillip Perron,

Levin, Lin, and Chu, are considered ineffective [45,46]. Therefore, our research utilizes the Pesaran CIPS test (2007) [47] since it can improve performance and efficiently address the difficulties associated with cross-sectional dependence.

The CIPS test is formulated as follows:

$$CIPS = \frac{1}{N} \sum_{i=1}^{N} t_i(N, T) \tag{6}$$

### 3.2.2. Implementing Westerlund's (2005) Approach

The test for stationarity alone does not provide sufficient evidence to establish the presence of a long-term relationship among the variables that have been chosen. To proceed, it is crucial to establish the presence of cointegration among the regressors. Although the utilization of cointegration is not a new idea, the contributions of Johansen (1988) [48], Pedroni (2001) [49], and Kao (1999) [50] have sparked renewed enthusiasm for this topic, particularly within the realm of panel data analysis. Nevertheless, it is crucial to acknowledge that these tests may not exhibit optimal performance in situations that involve cross-sectional dependence [51].

Given the circumstances, our research now focuses on the methodology proposed by Westerlund (2005) [46] since it is more appropriate for tackling challenges related to cross-dependence. This methodology provides effective outcomes not hindered by lingering dynamics, rendering it very helpful, even in scenarios involving a restricted sample size. The methodology proposed by Westerlund (2005) [52] is based on the utilization of two panel-specific autoregressive (AR) parameters.

(a) The panel specific A.R. test.

$$VR = \sum_{i=1}^{N} \sum_{t=1}^{T} \hat{E}_{it}^2 \hat{R}_i^{-1} \tag{7}$$

(b) The same A.R. test statistics.

$$VR = \sum_{i=1}^{N} \sum_{t=1}^{T} \hat{E}_{it}^2 \left( \sum_{t=1}^{n} \hat{R}_i \right)^{-1} \tag{8}$$

### 3.2.3. Driscoll and Kraay

The study applies the Driscoll and Kraay (DK) [53] method to estimate long-term impacts. This approach seems productive for analyzing panel data exhibiting geographical or temporal connections. DK is effective when there is a noticed spatial or temporal correlation within the dataset. Spatial dependency refers to the phenomenon where observations close to each other in space are more likely to exhibit correlation. On the other hand, temporal dependence refers to the concept that observations close to each other in time are more likely to display correlation. Considering panel data that potentially violates the assumption of independence can be beneficial. Incorporating spatial or temporal correlation can enhance the accuracy and efficiency of parameter estimates [54]. The above methods exhibit high flexibility and can be tailored to diverse datasets and correlation patterns.

## 4. Results

### 4.1. Spatial Graphing Assessment

Figure 2 shows the water usage efficiency of 29 provinces by year (2006–2020). It shows the average higher efficiency ranges between 1.01 and 1.23. The middle-efficiency ranges between 0.55–0.83. The lowest efficiency ranges between 0.21–0.49. Beijing (1.08), Shaanxi (1.01), Shanghai (1.23) and Tianjin (1.01) remained the higher efficient over the years. Six provinces (Guangdong, Shandong, Jiangsu, Inner Mongolia, Hebei and Zhejiang) are in the middle ranges (0.55–0.83). At the same time, nineteen provinces have the lowest

water usage efficiency (0.21–049). Qinghai and Ningxia are on the lowest rank (0.21) and (0.22) in water usage efficiency, respectively.

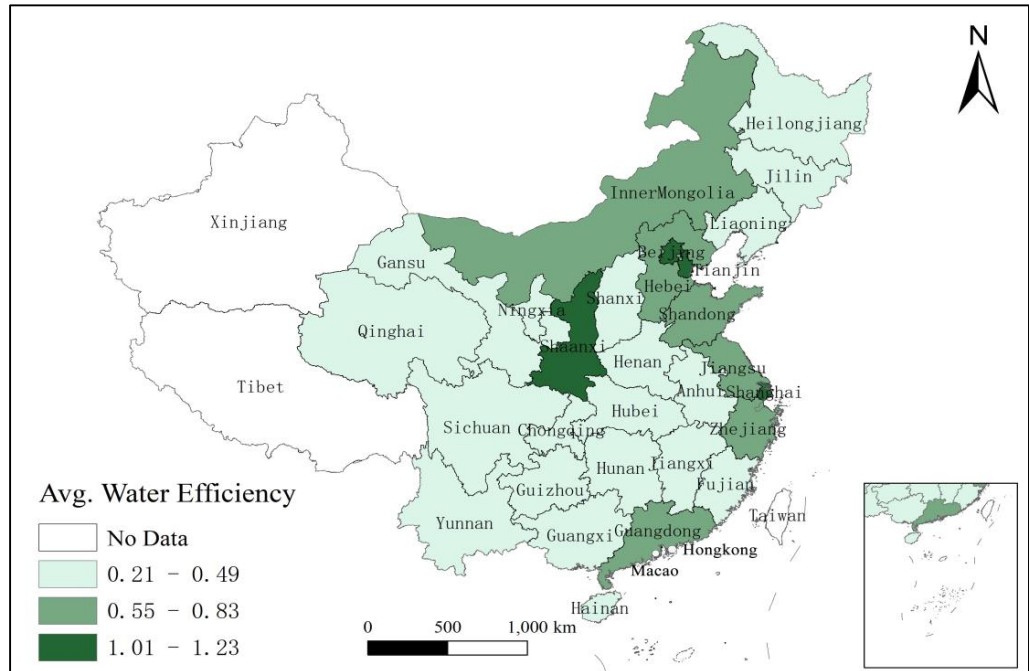

**Figure 2.** Water usage efficiency by year (2006–2020) of the provinces.

Figure 3 shows the average water usage efficiency by region; the eastern region has the higher efficiency with the value of (0.727). The central region shows less efficiency in water usage (0.383). However, the western region shows the middle range of efficiency. Overall, the regions are not efficient in water usage. It shows that China needs to reform and focus on strategies to improve water usage efficiency.

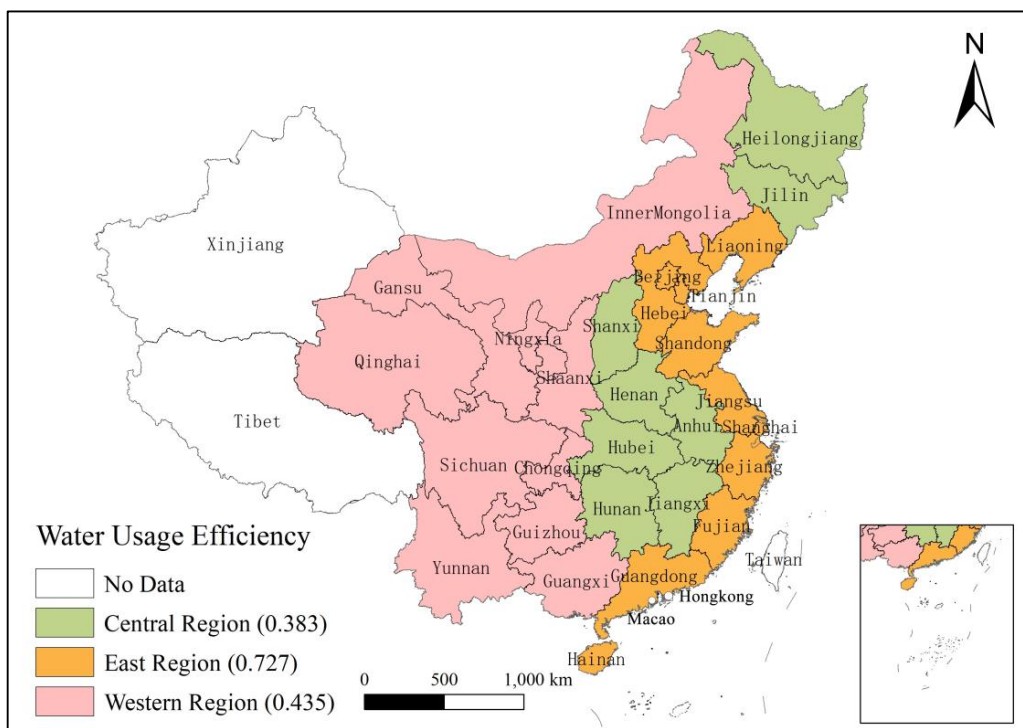

**Figure 3.** Water usage efficiency by regions (2006–2020).

Figure 4 shows the total water resources of all provinces from 2006–2020. Sichuan has the highest total water resources (37,690.11). Subsequently, Guangxi has the highest total water resources (29,394.47). In comparison, Beijing, Ningxia, Shanghai, Tianjin and Shanxi have less water resources. Beijing is situated in the northern area of China, characterized by an arid and semi-arid climate. Rainfall levels in this region are comparatively lower than in other regions of China, notably in the southern areas.

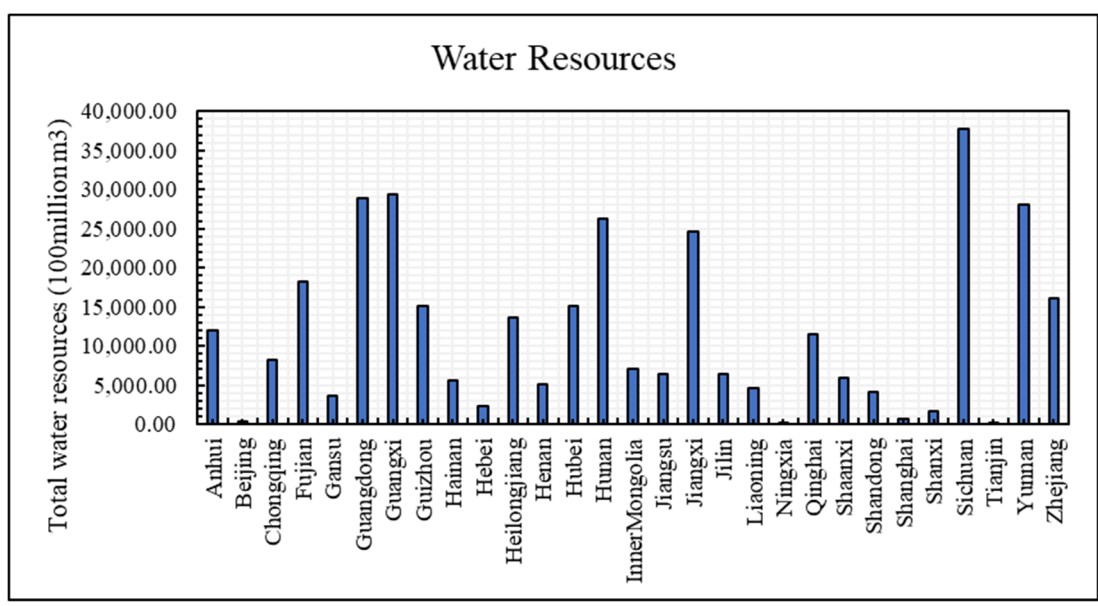

**Figure 4.** Water resources of the provinces (2006–2020).

The municipality's geographical positioning within a region characterized by aridity inherently constrains the availability of ample water supplies. The city of Beijing possesses a restricted number of natural water sources, such as rivers and lakes, within its territorial limits. However, in the water recycling Figure 5, Beijing is at the top in recycling water.

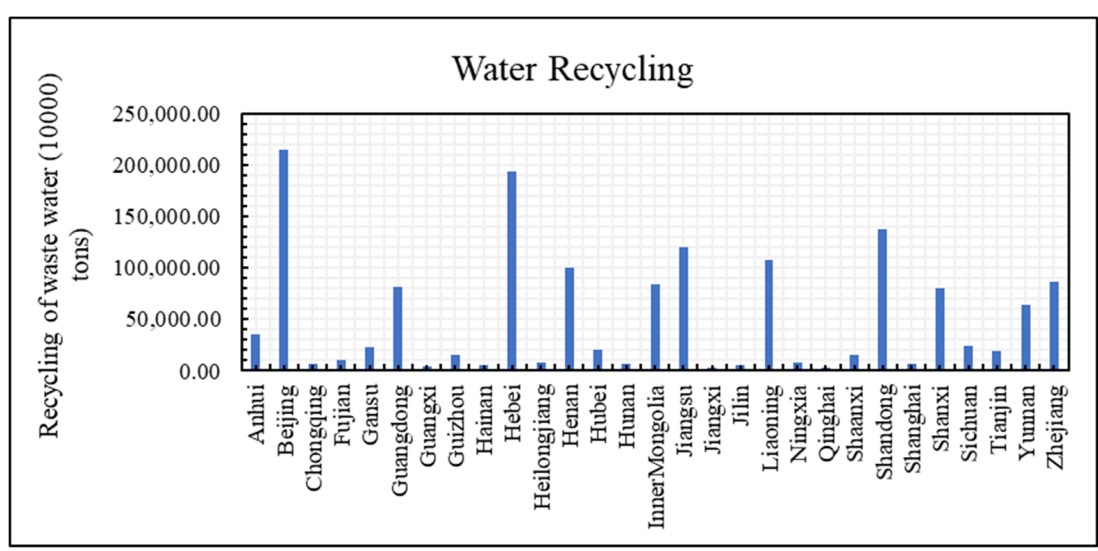

**Figure 5.** Water recycling of the provinces (2006–2020).

### 4.2. Empirical Findings and Discussion

The descriptive statistics (Table 1) provide valuable insights into various important aspects. The average water usage efficiency is moderate, with a mean value of 0.53. In contrast, the gross domestic product (GDP) has a comparatively high value of $50,208. The

availability of water resources demonstrates diversity among observations, with an average value of around 778.82. Water recycling exhibits significant variation, characterized by a mean value of 8545.81. The levels of sprinkling technology (technology for irrigation) also exhibit variation, as seen by a mean value of 121.23. There are significant variations in the presence and size of water reservoirs. The sizes of populations exhibit substantial variation, with an average of approximately 46.2 million. The secondary industry exhibits a mean value of 9392.62, whereas the education levels demonstrate considerable variation, with a mean value of 267,235.2. The statistics offer an initial comprehension of the dataset's primary tendencies and variations, serving as a foundation for subsequent analysis and decision-making in diverse domains such as economics, water technology, water resource management, technology, and education planning.

**Table 1.** Descriptive Summary.

| Variable(s) | Mean | Std. Dev. | Min. | Max. |
|---|---|---|---|---|
| WUEF | 0.5321333 | 0.2987302 | 0.1028 | 1.8176 |
| GDP | 50,207.69 | 54,882.67 | 5750 | 467,156 |
| WRS | 778.8157 | 720.1821 | 8.1 | 3237.3 |
| WR | 8545.81 | 13,590.28 | −3260 | 76,727 |
| SPR | 121.2325 | 246.4809 | 1.3 | 1661.17 |
| WTRS | 264.6713 | 234.1417 | 1.61 | 1263.89 |
| pop | $4.62 \times 10^7$ | $2.84 \times 10^7$ | 5,103,464 | $1.15 \times 10^8$ |
| Sind | 9392.617 | 8475.533 | 308.62 | 44,270.51 |
| EU | 267,235.2 | 171,134 | 29,313 | 749,826 |

Note: WUFE = water usage efficiency (caluted by DEA using Labor, capital stock, Water use, sewage andGDP), GDP is gross domestic output percapita, WRS = total water resources (100 million m$^3$), WR = Recycling of Wastewater (10,000 tons), SPR = Sprinkler Irrigation (1000 hectares), WTRS = water reservoirs in number, pop = population in total, Sind, secondary industry, EU is education measured by graduates secondary schools.

Table 2 displays the outcomes of a cross-sectional dependency examination conducted on multiple variables. The obtained *p*-values, all 0.000, provide substantial proof of the presence of cross-sectional dependency among the observations. This finding suggests that the variables under consideration are not independent of one another within the entire data set. The consistency of this dependency is highlighted by the consistent average joint T-statistic of 29.00 observed across all variables. Furthermore, the mean correlation coefficients (rho) exhibit a persistent positive trend, ranging from 0.84 to 1.00. This suggests a prevailing positive connection among the data for each variable. This suggests a relationship exists between changes in one variable and in other variables, emphasizing the importance of cross-sectional dependency in statistical analysis and modelling.

**Table 2.** Cross-sectional dependency.

| Variable(s) | CD-Test | *p*-Value | Average Joint T | Mean ρ | Mean abs(ρ) |
|---|---|---|---|---|---|
| WUEF | 46.077 | 0.000 | 29.00 | 0.84 | 0.84 |
| GDP | 53.538 | 0.000 | 29.00 | 0.97 | 0.97 |
| WRS | 51.83 | 0.000 | 29.00 | 0.94 | 0.94 |
| WR | 24.294 | 0.000 | 29.00 | 0.29 | 0.29 |
| SPR | 54.345 | 0.000 | 29.00 | 0.98 | 0.98 |
| WTRS | 50.029 | 0.000 | 29.00 | 0.91 | 0.91 |
| pop | 55.116 | 0.000 | 29.00 | 1.00 | 1.00 |
| Sind | 53.596 | 0.000 | 29.00 | 0.97 | 0.97 |
| EU | 52.526 | 0.000 | 29.00 | 0.95 | 0.95 |

Note: For variables description see note under Table 1.

The unit root analysis shows (Table 3) that "Water Usage Efficiency", "GDP", "Water Resources", "Recycling", "Irrigation Sprinkling Method", "Water Reservoir", "Population", "Secondary Industry", and "Education", are stationary after their first differences. This

means these variables are suitable for time series analysis and have no unit roots. For accurate and meaningful time series modelling, stationary data are needed to understand and predict economic and environmental patterns and relationships.

**Table 3.** Unit root Analysis.

| Variable(s) | CIPS (2007) | |
|:---:|:---:|:---:|
| | **Level** | **Firs-Diff** |
| WUEF | −4.949 *** | −6.097 *** |
| GDP | 0.3904 | −6.190 *** |
| WRS | −4.838 *** | −6.158 *** |
| WR | 0.8087 | −6.190 *** |
| SPR | 0.0104 | −5.663 *** |
| WTRS | 0.7583 | −5.5772 *** |
| pop | 1.4159 | −3.8266 *** |
| Sind | 0.5389 | −4.439 *** |
| EU | −1.6853 | −4.990 *** |

Note: For variables description see note under Table 1. *** $p < 0.01$.

Table 4 shows cointegration test results for two models, "Water Resources" and "Water Technology". Cointegration tests determine if variables move together across time. The "Water Resources" model has two test statistics. The first statistic, −1.9495, implies cointegration among variables in some panels (subsets of the data). This indicates long-term correlations between variables. The second statistic, −1.5521, shows weak cointegration in all panels. The *p*-values (0.0256 and 0.0603) reflect the significance of these results. The table shows two test statistics for the "Water Technology" model. The first, −1.5278, suggests panel cointegration. The second value, −1.5957, shows cointegration in all panels. Again, the *p*-values (0.0633 and 0.0553) show the significance of these findings. In conclusion, the "Water Resources" and "Water Technology" models show co-integration but with differing degrees of significance, showing that sets of variables within the panels are related across time. These correlations may vary in strength and importance across panels and models.

**Table 4.** Co-integration determination.

| Westerlund | | Statistic(s) | *p*-Value |
|:---|:---:|:---:|:---:|
| Water Resources Model | | | |
| some panels are cointegrated | Variance ratio | −1.9495 | 0.0256 |
| All panels are cointegrated | | −1.5521 | 0.0603 |
| Water Technology Model | | | |
| some panels are cointegrated | Variance ratio | −1.5278 | 0.0633 |
| All panels are cointegrated | | −1.5957 | 0.0553 |

For long-run assessment, we applied the Driscoll & Kraay. The results are described in Table 5. The study used three models: water-resources effects, water-saving-technology effects, and water resources-technology effects to assess the resource and technology impact. The primary focus of water resources (MD1) is to evaluate the effects of changes in water supplies on the efficiency of water usage. The observed coefficient of −0.0781 indicates a statistically significant negative relationship between the availability of water resources and water usage efficiency. This suggests that as the availability of water resources increases, there is a corresponding drop in water usage efficiency. From an economic perspective, it may be inferred that ample water resources could potentially reduce the motivation to adopt efficient water utilization strategies. This phenomenon may be attributed to the "tragedy of the commons" phenomenon when individuals or industries use a shared resource excessively when it is readily accessible without cost. In (MD2) we control the other economic effects to assess the dynamic impact of water resources on water usage efficiency.

**Table 5.** Water Resources-Water-saving-Technology Effects.

| | (MD1) | (MD2) | (MD3) | (MD4) | (MD5) | (MD6) | (MD7) | (MD8) |
|---|---|---|---|---|---|---|---|---|
| | **Water Resources Effects** | | **Water-Saving-Technology Effects** | | | | **Water Resources-Technology Effects** | |
| Variable(s) | **Dependent WUEF (Water Usage Efficiency)** | | | | | | | |
| WRS | −0.0781 *** | −0.0464 *** | | | | | −0.0524 *** | −0.00788 |
| | (0.00418) | (0.00711) | | | | | (0.0103) | (0.00986) |
| GDP | | −0.00509 | | −0.0510 *** | −0.0294 * | −0.710 *** | | −0.758 *** |
| | | (0.0166) | | (0.00511) | (0.0142) | (0.0669) | | (0.0958) |
| GDP$^2$ | | | | | | 0.0313 *** | | 0.0334 *** |
| | | | | | | (0.00293) | | (0.00435) |
| WR | | | 0.0203 * | 0.0415 *** | 0.0107 | 0.00938 | 0.00721 | 0.00571 |
| | | | (0.0106) | (0.00736) | (0.0102) | (0.0100) | (0.0123) | (0.00695) |
| SPR | | | | 0.0347 ** | 0.0175 ** | 0.0154 ** | 0.0168 *** | 0.00924 |
| | | | | (0.0126) | (0.00641) | (0.00714) | (0.00202) | (0.00909) |
| pop | | −0.0739 * | | | −0.188 *** | −0.202 *** | | −0.167 ** |
| | | (0.0397) | | | (0.0568) | (0.0561) | | (0.0713) |
| Sind | | 0.314 *** | | | 0.340 *** | 0.353 *** | | 0.295 *** |
| | | (0.0134) | | | (0.00993) | (0.00927) | | (0.0206) |
| EU | | 0.201 *** | | | 0.165 *** | 0.159 *** | | −0.0592 |
| | | (0.0337) | | | (0.0485) | (0.0481) | | (0.0550) |
| WTRS | | | | 0.0297 * | 0.0108 ** | 0.0381 *** | 0.0345 *** | 0.0779 *** |
| | | | | (0.0151) | (0.00426) | (0.00279) | (0.00303) | (0.00687) |
| Constant | 0.999 *** | 78.16 *** | 34.67 *** | 29.85 *** | 83.60 *** | 83.57 *** | 77.50 *** | 71.12 *** |
| | (0.0471) | (3.988) | (4.072) | (4.303) | (5.200) | (6.102) | (6.119) | (6.825) |
| Time Effect | Yes | Yes | Yes | Yes | Yes | Yes | Yes | Yes |
| Province Effect | Yes | Yes | Yes | Yes | Yes | Yes | Yes | Yes |
| Observations | 435 | 435 | 435 | 435 | 435 | 435 | 435 | 435 |
| Number of groups | 29 | 29 | 29 | 29 | 29 | 29 | 29 | 29 |

Note: Standard errors in parentheses *** $p < 0.01$, ** $p < 0.05$, * $p < 0.1$.

The water resources impact (−0.0464) remains negative, showing that Although individuals may have access to a greater quantity of water, they may not necessarily perceive the imperative to utilize it efficiently. Territories with considerable water resources may exhibit a reduced motivation to allocate significant resources towards developing and implementing efficient water infrastructure and management systems [55]. Cultural, economic, and legal issues affect how water resources and utilization efficiency relate in different places. Abundant water supplies managed efficiently and accompanied by educational and policy-driven measures to promote efficient use may minimize the negative link [56]. The following MD3-MD6 columns show the water-water-saving-technology effects. The impact of water-saving mechanisms on the efficiency of water usage seems positive. The opined coefficient of (0.0203) indicates a statistically significant positive relationship, implying that recycling technology significantly enhances China's water usage efficiency. It can be inferred that adopting such technology can enhance the efficiency of water usage, potentially lowering water-related expenses for both companies and homes [57,58]. The implementation of recycling technology contributes to enhancing water security by mitigating reliance on external water sources, particularly in locations that are susceptible to water scarcity [59,60].

Recycling water lessens demand on rivers, lakes, and aquifers [60]. Recycling treated wastewater for irrigation, industrial processes, and cooling systems reduces the requirement for fresh water. This conserves freshwater for drinking and cooking [61]. By adding water recycling, communities and enterprises become less dependent on one water source. Diversification strengthens droughts, water shortages, and other critical water supply disruptions. It ensures a more regular water supply, valuable for industries and agriculture. The utilization of recycling technology is under the fundamental tenets of a circular economy, which emphasizes the effective utilization, recycling, and utilization of resources, hence mitigating the necessity for fresh resource extraction and the development of trash. The use of circular water management practices facilitates the promotion of sustainability and the mitigation of environmental consequences. The increasing urban population in China demands recycling technology as a crucial component of urban water management. This technology can address growing urban water demand while mitigating the pressure on current water resources and infrastructure [62].

In the following columns (MD4-MD6), again, the observed coefficient shows the positive correlation between water recycling and water usage efficiency. The use of sprinkler systems plays a significant role in water usage efficiency. These advanced irrigation methods can increase the efficiency of water utilization. The positive coefficients in MD4-MD6 (0.0347, 0.0175, 0.0154) show a positive correlation between sprinkler systems irrigation strategies and heightened water efficiency. The potential reason for this phenomenon can be attributed to these methodologies' enhanced accuracy and control, resulting in decreased water consumption [63,64]. This practice effectively mitigates water loss caused by runoff and evaporation, hence enhancing the efficiency of water delivery to plants in the areas where it is most required. Irrigation sprinkling reduces waste, conserves water, and promotes ecologically sound farming by precisely targeting and monitoring the water supply to crops. The third important water-saving component is the water reservoirs.

The impact of water reservoirs is positive to increase the water usage efficiency. Water reservoirs can potentially improve the efficiency of water usage by providing a consistent and dependable water supply. Moreover, they can mitigate the effects of seasonal fluctuations, facilitate hydropower generation, enhance ecosystems, provide recreational activities, act as a contingency water source during emergencies, and alleviate the strain on groundwater resources [65]. Reservoirs are crucial in serving as a vital emergency water supply during catastrophes, offering drinkable water to impacted communities where alternative supplies may be disrupted [66]. The last two columns describe the combined effects of water resources and technology with the controlled parameters. The results show that water technology is more efficient than water resources to increase water usage efficiency. The impact of economic development (GDP) on water usage efficiency is negative throughout the regressions. It implies that during the early stages of economic development, there is a possibility for increasing demand for water-intensive activities such as industrial production and agriculture, which can potentially lead to a decline in water efficiency. However, the positive coefficient for the square of GDP (0.0313, 0.0334) suggests that as GDP increases, the rate of decrease in water usage efficiency slows down. The findings suggest that once economies attain a specific development threshold, they allocate greater resources towards implementing water conservation and efficiency measures.

In all models, it is observed that population has a detrimental effect on water usage efficiency. As the population grows, there is a tendency for water efficiency to decline. From an economic perspective, there is a correlation between increased water consumption for home and industrial purposes in densely populated regions, which can exert pressure on water supplies. Both factors have positive coefficients, suggesting increased secondary industry and education involvement may enhance water usage efficiency. This implies that implementing economic diversification and education initiatives can potentially result in adopting more sustainable water management methods.

Table 6 shows the synergy of water resource agglomeration and conservation technologies' impact on water usage efficiency. MD1 coefficient of −0.0832 implies that water usage efficiency decreases with water resource increase. This suggests that the availability of abundant water supplies may reduce the incentive to adopt optimal water utilization practices. Similarly, recycling and sprinkling have a positive impact, indicating that adopting recycling practices is associated with higher water usage efficiency. In MD1-MD2 and MD3, the study uses the mediating impact of different methods and technology with water resources. The findings indicated that the interaction terms of recycling, sprinkling, and reservoirs with water resources are statistically significant, indicating that the relationship between water resources and water usage efficiency depends on the other factors. Just abundant water resources are not enough if it is not managed efficiently. Other economic factors, such as population economic development, can increase the demand for water and put pressure on the water resources. So, the provinces need to manage them efficiently to meet the requirements. Education can play an essential role as an awareness tool among the people and use the water efficiently.

**Table 6.** Moderation Effects of Conservation Water Technologies with Water Resources.

| Variable(s) | (MD1) | (MD2) | (MD3) |
|---|---|---|---|
| | Dependent WUEF (Water Usage Efficiency) | | |
| WRS | −0.0832 ** | −0.116 ** | 0.0682 |
| | (0.0303) | (0.0558) | (0.120) |
| GDP | −2.098 *** | −1.884 ** | −0.788 * |
| | (0.715) | (0.734) | (0.399) |
| GDP$^2$ | 0.0922 *** | 0.0814 ** | 0.0348 * |
| | (0.0324) | (0.0333) | (0.0174) |
| WR | 0.264 *** | | |
| | (0.0647) | | |
| SPR | | 0.0311 *** | |
| | | (0.00513) | |
| pop | −0.00739 | −0.307 *** | −0.161 |
| | (0.144) | (0.0463) | (0.145) |
| Sind | 0.270 *** | 0.237 *** | 0.298 *** |
| | (0.0626) | (0.0462) | (0.0433) |
| EU | 0.281 ** | 0.0968 *** | 0.0305 *** |
| | (0.127) | (0.0333) | (0.00548) |
| WTRS | | | 0.300 *** |
| | | | (0.0438) |
| WRS×WR | 0.0923 ** | | |
| | (0.0358) | | |
| WRS × SPR | | 0.0853 * | |
| | | (0.0440) | |
| WRS × WTRS | | | 0.0307 *** |
| | | | (0.00538) |
| Constant | 12.68 *** | 26.65 ** | 48.20 *** |
| | (3.981) | (10.19) | (3.023) |
| Time Effect | Yes | Yes | Yes |
| Province Effect | Yes | Yes | Yes |
| Observations | 435 | 435 | 435 |
| Number of groups | 29 | 29 | 29 |

Note: Standard errors in parentheses *** $p < 0.01$, ** $p < 0.05$, * $p < 0.1$.

## 5. Conclusions

In an era where the sustainable management of water resources has become an imperative global concern, China has also decided to improve its water utilization efficiency through innovative methods and reforms. Therefore, the effective management and efficient utilization of water resources in China have emerged as significant focal points to mitigate water-related issues. Thus, this study is developed to explore the synergy of water resource agglomeration and innovative conservation technologies on the water usage efficiency at the province and regional levels in China from (2006–2020).

In the first stage, the study employs the super SBM-DEA approach to analyze the water usage efficiency of the province and regions. SBM-DEA analysis revealed that Beijing (1.08), Shaanxi (1.01), Shanghai (1.23) and Tianjin (1.01) remained the higher efficient over the years. Six provinces (Guangdong, Shandong, Jiangsu, Inner Mongolia, Hebei, and Zhejiang) are in the middle ranges (0.55–0.83). In comparison, nineteen provinces have the lowest water usage efficiency (0.21–049). Qinghai and Ningxia are on the lowest rank (0.21) and (0.22), respectively.

In the second stage, we find the dynamic nexuses between water resources, water technologies and water usage efficiency by applying a systematic econometric series. The findings recommended that the water resources impact remains negative, showing that Although individuals may have access to a greater quantity of water, they may not necessarily perceive the imperative to utilize it efficiently. Territories with considerable water resources may exhibit a reduced motivation to allocate significant resources towards developing and implementing efficient water infrastructure and management systems.

The impact of water-saving mechanisms on the efficiency of water usage seems to be positive. The opined coefficient of recycling technology significantly enhances the water usage efficiency in China's province. It can be inferred that adopting such technology has the potential to enhance the efficiency of water usage.

Recycling treated wastewater for irrigation, industrial processes, and cooling systems reduces the requirement for water. The use of sprinkler systems plays a significant role in water usage efficiency. These advanced irrigation methods can increase water utilization efficiency in China. Water can mitigate the effects of seasonal fluctuations, facilitate hydropower generation, enhance ecosystems, provide recreational activities, act as a contingency water source during emergencies, and alleviate the strain on groundwater resources. One important finding is that the impact of economic development (GDP) on water usage efficiency is negative during the early stages of economic development; there is a possibility for increasing demand for water-intensive activities such as industrial production and agriculture, potentially leading to a decline in water efficiency. However, as GDP increases, the decrease in water usage efficiency slows down. The findings suggest that once economies attain a specific development threshold, they allocate greater resources towards implementing water conservation and efficiency measures. The increasing population stressed the water demand and sewage and decreased water usage efficiency. However, education can be a way to increase awareness and skills to improve water utilization. The secondary industry also seems to be effective in increasing water efficiency. The findings indicated that the interaction of recycling, sprinkling, and reservoirs with water resources is statistically significant, indicating that the relationship between water resources and water usage efficiency depends on the other factors. Just abundant water resources are not enough if it is not managed efficiently.

To summarize, the mere availability of abundant water resources does not guarantee efficient utilization. Instead, adopting water-saving mechanisms, recycling technologies, and advanced irrigation methods are important to achieving efficient water resource management in China.

The study's findings lead us to several policy recommendations to improve water resource management and efficiency in China. China should promote recycling treated wastewater for irrigation and industry. Enhancing water efficiency in regions with limited water resources, such as Ningxia, can be achieved through developing water recycling technology, modernizing irrigation systems, implementing integrated water resource management, and adopting tiered water pricing schemes. A tiered water pricing system employs multiple price levels, wherein the initial tier is characterized by lower rates aimed at addressing fundamental family water requirements. As water consumption increases, further tiers are introduced with incrementally increasing charges per unit, promoting a water conservation culture. Thus, through this system, everyone can have access to essential water services at a fair price and discourage wasteful use to encourage conservation, especially in drought-stricken areas. Further, develop strong water monitoring and data collecting systems to effectively evaluate and track water use, accessibility, and quality.

The comprehensive approach to efficient water resource management in China is achieved by the active involvement of local communities in decision-making processes and conservation initiatives. It is imperative to implement educational initiatives and raise awareness regarding water conservation. Additionally, enhancing governance and regulation and allocating resources towards research and innovation are of utmost importance. Promoting sustainable economic development, considering the implications of population expansion in urban design, and enhancing reservoir management are crucial measures. In addition to being vital, capacity building, cross-border collaboration on transboundary water management, and long-term climate change planning are crucial.

Future research endeavors in the field of water resource management in China ought to prioritize the comprehensive evaluation of the enduring effects of policy interventions and the adoption of technological advancements while considering the dynamic interplay of environmental and socio-economic elements. Furthermore, research must thoroughly

investigate climate change adaptation techniques to optimize water resource allocation and enhance infrastructure management efficiency.

**Author Contributions:** Conceptualization, R.Y.; methodology, R.Y. and W.U.H.S.; software; R.Y. and W.U.H.S.; validation, R.Y. and W.U.H.S.; formal analysis, R.Y. and W.U.H.S.; investigation, R.Y. and W.U.H.S.; resources, R.Y. and W.U.H.S.; data curation, R.Y. and W.U.H.S.; writing—original draft preparation, R.Y.; writing—review and editing, G.H.; Y.Y. and C.T.; visualization, G.H.; supervision, G.H. All authors have read and agreed to the published version of the manuscript.

**Funding:** This study was supported in part by the National Social Science Foundation, P.R. China (Project No. 20BJY087). 2022 Humanities and Social Sciences Research Project of the Chinese Ministry of Education—Youth Fund Project (No. 22YJCZH069).

**Data Availability Statement:** National Bureau of Statistics of China; Ministry of Environmental Protection of China; China Agricultural Machinery Industry Yearbook.

**Conflicts of Interest:** The authors declare no conflict of interest.

## Appendix A

**Table A1.** China Province and Regional Distribution.

| Central Region | Beijing | Western Region |
|---|---|---|
| Anhui | Fujian | Chongqing |
| Heilongjiang | Guangdong | Gansu |
| Henan | Hainan | Guangxi |
| Hubei | Hebei | Guizhou |
| Hunan | Jiangsu | Inner Mongolia |
| Jiangxi | Shandong | Ningxia |
| Jilin | Shanghai | Qinghai |
| Shanxi | Tianjin | Shaanxi |
| Yunnan | Zhejiang | Sichuan |
| **Eastern Region** | Liaoning | |

**Table A2.** Variables and data source.

| Variables | Units | Data Sources |
|---|---|---|
| Water Usage Efficiency | Inputs: Labor, capital stock, Water use. Bad output: sewage. Desired output: GDP | |
| GDP | Gross domestic output per capita | |
| Water Resources | Total Water Resources (100 million m$^3$) | National Bureau of Statistics of China |
| Recycling | Recycling of Wastewater (10,000 tons) | Ministry of Environmental Protection of China |
| Irrigation Sprinkling | Sprinkler Irrigation (1000 hectares) | China Agricultural Machinery Industry |
| Water Reservoir | Number of reservoirs | Yearbook |
| Population | Total population | |
| Industry | Secondary industry | |
| Education | Graduates Secondary Schools | |

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
