# Peer review of "The Synergy of Water Resource Agglomeration and Innovative Conservation Technologies on Provincial and Regional Water Usage Efficiency in China: A Super SBM-DEA Approach"

_water, doi:10.3390/w15193524_

Round 1

Reviewer 1 Report

Page 3, Lines 135 – 136: “Forth, education impact on increasing water usage efficiency.” This is an incomplete sentence.

Page 3, Lines 149 – 151: A “rebound impact” is mentioned in these lines of text. What exactly is a “rebound impact”? It would be good to define this for the reader.

Page 4, Spatial graphing assessment section and Figures 1, and 2: The authors report water usage efficiencies by both province and region in these two figures, but they don’t site where these efficiencies come from. Are they from another study? Are these the water usage efficiencies estimated by the data envelopment analysis used in this study? If these are the efficiencies estimated by the present study, they need to be reported in the Results section of the manuscript rather than after the literature review. It is not appropriate to report results before explaining how these results are arrived at.

Page 6, Figures 3 and 4: The authors report water resources by province in Figure 3 and water recycling by province in Figure 4. What units are the water resources in Figure 3 measured in? What units are the water recycling numbers in Figure 4 measured in? How are each of these variables defined? What is the source of these data? Both figures need an axis title to describe the units that water resources/water recycling are measured in.

Pages 10 – 11, Tables 1, 2, and 3: Variables are reported as acronyms or abbreviations in these tables, but nothing is known about what these acronyms or abbreviations mean or the units in which each of these variables are measured. A footnote is needed at the end of each table to spell out each variable and provide the units for which each variable is measured. Tables should stand alone without reference in the text.

Page 18, Line 526: The authors mention the introduction of tiered water pricing schemes as a policy recommendation for improving water usage efficiencies in China. How can these enhance water usage efficiencies? A bit more information needs to be provided by the authors about tiered water pricing schemes.

Author Response

we have incorporated the all comments please find the attached file

Author Response

We have tried to incorporated the comments and improved the revised version.

Reviewer 3 Report

The paper is about the synergy of water resource agglomeration and innovative conservation technologies on provincial and regional water usage efficiency in China. Interesting and needed as the study concept is understudied in the region. Overall, the paper needs some significant improvements/revisions before its possible publication. Below are some of my concerns to be addressed:

Though terms are used interchangeably, I suggest being uniform (Consistency) throughout the whole of the manuscript. Choose one word/concept to be used. This will help you in the improvement of the manuscript to avoid further confusion.

The literature review is somehow weak (Improvements needed). By reading through, it is difficult to grasp the key justification for the need of this research. The manuscript needs to clearly elaborate more and show, what are the problems with the existing works in the field either globally, regionally, or nationally? Without this, readers would have difficulties in seeing the merit of this paper. Toward the end of the introduction section, the author (s) are requested to show what is missing in these previous works to grasp the real limitations of these studies leading to the motivation of conducting this specific research. Otherwise, it is questionable/dubious when the novelty is considered, authors must underline and stress on the novelty of the paper.

State the specific objectives/Aims of your study. Please see papers, optimal development of agricultural sectors in the basin based on economic efficiency and social equality and optimal design and operation of a hydropower reservoir plant using a WEAP-Based simulation–optimization approach for more literature.

The designed flowchart is necessary to really clear to describe the steps of the model and the methodology in the manuscript

While the result section is well written, there is limited discussion about this study. This makes the whole part of the discussion weak and poor. Comparing your results is not just enough but also should consider providing the implication of your findings. The author (s) are requested to dig deep into the recent literature (Consulting recent publications) on the topic to discuss the overall results of the study.

The authors have succinctly summarized the major findings but toward the end, the significance of the research findings was not provided. The major weakness of this section is that there is a lack of concluding remarks based on your findings. Recommendations for future direction/orientation for further research based on the remaining gaps is highly encouraged.

Re-check the journal guidelines for authors to update their references (Cited sources). Revisit long sentences and try to shorten them so that readers of the manuscript cannot get lost.

Author Response

(The authors gave the same response as above.)

Round 2

Reviewer 2 Report

Whilst I still have some disagreements with the authors (eg the distinction between recycling and reuse, the adequacy of sprinklers versus drip irrigation), these are not sufficient to say that there are any significant problems.  It is a greatly improved paper which is publishable as it now stands.

Reviewer 3 Report

The paper is well revised. Please  improve quality of figures during publication process.